# SDF-1α-Releasing Microspheres Effectively Extend Stem Cell Homing after Myocardial Infarction

**DOI:** 10.3390/biomedicines11020343

**Published:** 2023-01-25

**Authors:** Karolina Bajdak-Rusinek, Agnieszka Fus-Kujawa, Piotr Buszman, Dorota Żyła-Uklejewicz, Katarzyna Jelonek, Monika Musiał-Kulik, Carlos Fernandez, Magdalena Michalak, Kurian George, Janusz Kasperczyk, Paweł Buszman

**Affiliations:** 1Department of Medical Genetics, Faculty of Medical Sciences in Katowice, Medical University of Silesia, 40-752 Katowice, Poland; 2Cardiology Department, Andrzej Frycz Modrzewski Krakow University, 30-705 Krakow, Poland; 3Center for Cardiovascular Research and Development, American Heart of Poland, 40-028 Katowice, Poland; 4Centre of Polymer and Carbon Materials, Polish Academy of Sciences, 41-819 Zabrze, Poland; 5Students Scientific Society, Faculty of Medical Sciences in Katowice, Medical University of Silesia, 40-055 Katowice, Poland; 6Department of Epidemiology, Medical University of Silesia, 40-055 Katowice, Poland

**Keywords:** SDF-1α, bone marrow mesenchymal stromal cells (bmMSCs), biodegradable polymeric microspheres, myocardial infarction

## Abstract

Ischemic heart disease (IHD) is one of the main focuses in today’s healthcare due to its implications and complications, and it is predicted to be increasing in prevalence due to the ageing population. Although the conventional pharmacological and interventional methods for the treatment of IHD presents with success in the clinical setting, the long-term complications of cardiac insufficiency are on a continual incline as a result of post-infarction remodeling of the cardiac tissue. The migration and involvement of stem cells to the cardiac muscle, followed by differentiation into cardiac myocytes, has been proven to be the natural process, though at a slow rate. SDF-1α is a novel candidate to mobilize stem cells homing to the ischemic heart. Endogenous SDF-1α levels are elevated after myocardial infarction, but their presence gradually decreases after approximately seven days. Additional administration of SDF-1α-releasing microspheres could be a tool for the extension of the time the stem cells are in the cardiac tissue after myocardial infarction. This, in turn, could constitute a novel therapy for more efficient regeneration of the heart muscle after injury. Through this practical study, it has been shown that the controlled release of SDF-1α from biodegradable microspheres into the pericardial sac fourteen days after myocardial infarction increases the concentration of exogenous SDF-1α, which persists in the tissue much longer than the level of endogenous SDF-1α. In addition, administration of SDF-1α-releasing microspheres increased the expression of the factors potentially involved in the involvement and retention of myocardial stem cells, which constitutes vascular endothelial growth factor A (VEGFA), stem cell factor (SCF), and vascular cell adhesion molecules (VCAMs) at the site of damaged tissue. This exhibits the possibility of combating the basic limitations of cell therapy, including ineffective stem cell implantation and the ability to induce the migration of endogenous stem cells to the ischemic cardiac tissue and promote heart repair.

## 1. Introduction

Cardiovascular diseases (CVD) development due to atherosclerosis constitutes a main cause of death in the world. Ischemic heart disease (IHD), together with its complications, such as myocardial infarction (MI) and heart failure (HF), are at the forefront of CVDs and present a serious problem from a social and economic point of view. Moreover, the risk of these diseases increases with age; thus, light of the demographic projections, an increase in the number of CVD patients is to be expected due to the ageing population [1,2,3].

The current methods for treating ischemic heart disease include the modification of risk factors, pharmacotherapy, and interventional treatment. Technological progress has also contributed to the significant reduction of mortality rates in patients with myocardial infarction. On the other hand, in the long term, the incidence of heart failure as a result of post-infarction myocardial remodeling (CM) is steadily increasing [4]. Modern medicine faces an enormous challenge, which requires innovative solutions adapted to the new requirements.

There is increasing evidence that one naturally occurring process is the mobilization of stem cells to the heart and differentiation into cardiac myocytes. However, it is a slow progression for the significant recovery of left ventricular function following myocardial infarction [5,6]. Therefore, one of the most promising strategies to aid faster and more efficient recovery in the treatment of cardiovascular diseases is stem cell therapy. Although it is a safe therapy, there are associated limitations, mainly concerning the low efficacy of stem cell homing in the myocardium [7,8]. Previously, direct injection of stem cells into the ischemic heart specimens failed to improve cardiac function, because the microenvironment of the ischemic myocardium did not promote exogenous cell survival, differentiation, and integration into the recipient heart [9]. As a result, in recent years, emphasis has been placed on optimizing cell therapy by mobilizing endogenous stem cells into the ischemic heart, as well as on the method of administration and duration of the treatment.

Chemokine SDF-1 (stromal cell-derived factor 1) is a novel candidate to mobilize stem cell homing to the ischemic heart [10,11]. It is known that, after MI, intravenously delivered stem cells localize at the peri infarction area, suggesting the presence of local chemotactic factors, including SDF-1 [12]. Although the levels of endogenous SDF-1 are elevated after MI, their presence decreases gradually after 4–7 days. From a practical point of view, it is advisable to take steps to obtain high concentrations of SDF-1 in the heart [12,13].

Herein, we show that the controlled release of SDF-1α from biodegradable microspheres into the pericardial sac increases the recruitment of stem cells to the heart after MI, and also increases stem cell homing (Figure 1).

## 2. Methods

### 2.1. Animal Studies

This part of the project was performed at the Center for Cardiovascular Research and Development of American Heart of Poland. All procedures were approved by the Animal Ethics Committee (Contract No. 64/2018, 19/2021). All animals received standard care in accordance with the Animal Welfare Act and the “Guide for the Care and Use of Laboratory Animals”. A total of 33 domestic swine (*Sus scrofa domesticus*), with an average weight of 40 kg, were incorporated in this study.

### 2.2. Animal Experimental Design

The detailed protocol of a closed thoracic model of myocardial infarction/reperfusion was previously published by our group [14,15,16]. After the appropriate depth of anesthesia was achieved under sterile conditions, a percutaneous vascular sheath (6F) was placed in the femoral artery for arterial access. Using standard percutaneous coronary intervention techniques, a 6F JR3.5 guide catheter was inserted into the left coronary artery, and initial cine angiography was recorded with manual injections of radiographic contrast agent. A simulated myocardial infarction in a pig consisted of a balloon blocking the left anterior descending artery for 60 min, which resulted in transmural necrosis in the area it supplies (a recognized model of myocardial infarction) [17]. The procedure was completed 30 min after the blood flow was restored, then a control coronary angiography was performed to confirm the patency of the artery. In the test group, microspheres (MS) were delivered to the pericardial sac 14 days after the myocardial infarction. Control animal subjects did not receive SDF-1α-released microspheres. The animals were sacrificed 24 h, 72 h, 7 days, and 14 days after MI and 24 h, 72 h, 7 days, 14 days, 3 weeks, 9 weeks, and 15 weeks after microspheres delivery. The procedure was performed by qualified personnel under general anesthesia by intravenous injection of a commercial euthanasia solution (Figure 2).

Subsequently, analysis of the SDF-1α concentration in the damaged tissue was carried out using ELISA, and gene expression was analyzes using RT-qPCR. The study procedure flowchart is presented in Table 1.

### 2.3. Microspheres with SDF-1α

The microspheres (MS) were prepared from poly(L-lactide/glycolide/trimethylene carbonate) (PLA/GA/TMC) according to the previously reported procedure [18]. For a short period of time, the water/oil/water (w/o/w) emulsion method was used to produce the microspheres. SDF-1α was mixed with bovine serum albumin (BSA) (1:9 *w*/*w*) and dissolved in deionized H_2_O (3.5% *w*/*v*). The polymer was dissolved in dichloromethane (12.5% *w*/*v*). The w/o phase was sonicated for 15 s (Hielscher UP200Ht, Teltow, Germany) and added dropwise to 5% polyvinyl alcohol (PVA) (Kinematica, Polytron PT 2500 E, Malters, Switzerland) at 13,000 rpm. The resulting emulsion was stirred (100 rpm) overnight for solvent evaporation, and the MS were collected by centrifugation (Eppendorf 5810R, Darmstadt, Germany). The obtained MS were then freeze-dried (Christ, Alpha 1-2 LD plus, Osterode am Harz, Germany) and stored at 4 °C 

### 2.4. Enzyme-Linked Immunosorbent Assay (ELISA)

Tissues were harvested and stored at −80 °C. At the indicated time points they were thawed on ice and rinsed with cold Dulbecco’s Phosphate Buffered Saline (DPBS) (PAN Biotech, Aidenbach, Germany). The tissues were then cut into small pieces, weighed, and submersed in an appropriate amount of RIPA Buffer (Abcam, Boston, MA, USA) containing a protease inhibitor (Roche, Basel, Switzerland). Subsequently, they were blended at high speed until completely homogenized using a homogenizer (Unidrive × 1000D CAT, Ballrechten-Dottingen, Germany). Thereafter, the homogenates were centrifuged twice at 12,000 rpm in 10 min, and the supernatants were collected. The concentration of exogenous and endogenous SDF-1α was measured using an ELISA kit (Abclonal, Woburn, MA, USA).

### 2.5. RNA Isolation and Quantitative RT-PCR

Total RNA was isolated using the RNeasy Fibrous Tissue Mini Kit (Qiagen, Hilden, Germany) according to the manufacturer’s protocol. cDNA was synthesized from 1 µg RNA with the Revert Aid First Strand cDNA Synthesis Kit (Thermo Scientific, Karlsruhe, Germany) according to the manufacturer’s instructions. Relative expression levels were measured in triplicate in a Roche Light Cycler 480 using Power SYBR Green PCR Master Mix (Applied Biosystems, Darmstadt, Germany), 300 mM primers (Appendix A), and 1/15 cDNA stock. Relative expression levels were calculated and normalized to those of GAPDH, applying the Pfaffl method [19].

### 2.6. Statistical Analysis

Statistical analyses of the data were performed with Microsoft Excel software. Normalized relative expression levels were used to calculate the mean and the SD of all experiments (represented by columns and error bars in the figures). 

## 3. Results

### 3.1. Intrapericardial Administration of Microspheres Increases the Concentration of SDF-1α in the Myocardium

In our previously conducted work, we presented that the microspheres we obtained were capable of the controlled and sustained release of SDF-1 in vitro. Moreover, the SDF-1α factor released from the microspheres was able to stimulate the migration of bmMSCs [18].

In the present study, we wanted to evaluate the long-term biological effect of SDF-1α released from bioresorbable microspheres into the pericardial sac. Therefore, two weeks after recovery from MI, animals were subjected to intra-pericardial delivery of SDF-1α-releasing microspheres, and quantification of endogenous porcine SDF-1α and exogenous (released from the microspheres) levels of SDF-1α were performed at appropriate post-infarction time points.

The analysis showed that the endogenous levels of SDF-1α increases after the myocardial infarction, reaching the highest level on the third day after the MI. Its level then gradually decreases (Figure 3A). In turn, the concentration of exogenous SDF-1α remained at a very high level, even for 15 weeks after the intrapericardial administration of SDF-1α-releasing microspheres (Figure 3B).

In conclusion, pharmacokinetic studies confirmed the presence of SDF-1α in the myocardium after intrapericardial administration of microspheres by ELISA. Moreover, the concentration of exogenous SDF-1α persisted in the tissue much longer than the level of endogenous SDF-1α.

### 3.2. SDF-1α-Releasing Microspheres Affect Gene Expression in the Heart after MI

The following task was the analysis of the expression of the genes responsible for migration and for the stem cells involved at the site of the damaged tissue of the heart muscle in response to SDF-1α. The infarct tissues of animals that received SDF-1α-releasing microspheres into the pericardial sac 14 days after the MI (study group) was compared with those that did not receive SDF-1α-releasing microspheres (control group) (Figure 4). In the control group, the levels of SDF-1α and C-X-C chemokine receptor type 4 (CXCR-4) doubled after 72h and then began to decline, as seen on day 14 after MI. Additionally, the levels of VEGFA and VCAM increased slightly, and after two weeks their decrease was observed. In contrast, an increase in stem cell factor (SCF) expression was not observed. In turn, in the study group, by administering SDF-1α-releasing microspheres, a prolonged effect on the expression of the analyzed genes was observed. SDF-1α, CXCR-4, VCAM, VEGF, and SCF were significantly elevated, which was observed even 15 weeks after the MI.

The results obtained indicate that by the additional administration of SDF-1α-releasing microspheres, the time of stem cell homing in tissue after myocardial infarction can be effectively extended. This, in turn, could constitute a novel therapy for more efficient regeneration of the heart muscle after injury.

## 4. Discussion

Myocardial infarction is a pathological process characterized by necrosis of the cardiac tissue as a result of persistent ischemia [20]. In recent years, the view on the mechanism of action of stem cell therapy in the treatment of this disease has evolved [21]. It has not yet been confirmed that the human myocardium regenerates, but the observed effects are believed to be related primarily to the paracrine effect leading to the formation of new vessels (angiogenesis), recruitment of resident cardiac stem cells, and inhibition of cardiomyocyte apoptosis. One of the key elements determining the myocardial repair response associated with stem cell therapy are endogenous chemotactic mechanisms occurring in the cardiac muscle, which are stimulated in response to ischemia, and condition temporary or permanent homing by exogenous stem cells in damaged tissue. Such mechanisms include increased expression of cytokines, chemokines, and growth factors at the site of tissue damage, and the presence of receptors for these factors on the stem cells homing on the bone marrow and circulating in the blood [11,22,23].

Among the signaling axes (factor/receptor) involved in the process of mobilization and migration of the stem cells, the most important ones have been noted to be SDF-1-CXCR4 [24], hepatocyte growth factor (HGF)-c-Met [25], SCF-C-kit [26], and VEGF-VEGF receptor (VEGFR) [27], which are particularly important in the regeneration of endothelial cells. SDF-lα occurs in high concentrations in the bone marrow, where it is produced by stromal cells such as osteoblasts, endothelial cells, and reticular cells. The SDF-1/CXCR4 axis stimulates migration and nesting of the cells in bone marrow niches and conditions the mobilization of the stem cells to the peripheral blood [28]. 

As one of the first investigations undertaken, the team confirmed the presence of CXCR4+, C-kit+, and C-met+ cells circulating in the blood of patients who suffered from myocardial infarction. Importantly, the cells with a receptor for SDF-lα show the increased expression of genes typical of cardiomyocytes and endothelial cells [29]. Furthermore, as shown in the study on the myocardial infarction mouse model, the cells migrating to the SDF-lα gradient show a notably increased activity of early transcription factors for cardiac cells, which suggests their essential role in repair processes [11]. The findings of experimental studies and biopsies of the human myocardium provide convincing information to confirm the increased production of SDF-lα within a few hours after ischemia in the peri-infarct zone, which were reported to promote the nesting of the heart tissue committed stem cells [30]. This mechanism possibly conditions the gathering of cells primarily in the peri-infarct zone after the intracoronary administration [31]. As established, directly after infarction, bone marrow cells show a low chemotactic response, which increases significantly only 4–7 days after infarction. At that time, the expression of SDF-1 in the myocardium is reduced, as compared to the value just after infarction. This leads to the maximum chemotactic response of the CXCR4+ cells to the gradient of SDF-1 concentrations occurring naturally much later in relation to the peak expression of SDF-1 [12]. This is confirmed by our results, where it can be observed that the endogenous level of SDF-1α in the infarcted tissue reaches the highest level on the 3rd day after the MI, and then its level gradually decreases (Figure 3A). However, the intensified migration of the stem cells according to the SDF-1 gradient is one of the prognostic factors behind effective stem cell therapy (REPAIR-AMI) [18,32]. As revealed in experimental studies, the concentrations of SDF-1 in the blood are much lower than in the bone marrow, which conditions the retention of such cells. In such concentrations, circulating SDF-1 does not induce any significant migration of the stem cells. Obtaining concentrations higher than physiological ones leads to a significant increase in migration, which indicates that the local concentration of SDF1 in the heart should be very high. 

Currently, several methods of increasing SDF-1 concentrations in tissues are being investigated as potential therapeutic methods. However, there are some limitations, including the rapid diffusion of SDF-1 and its inactivation by proteases, especially in recent ischemia.

This can be counteracted by modifying the structure of SDF-1, which makes the chemokine less susceptible to the action of proteolytic enzymes (S4V) [33], or the sustained release of a biocompatible carrier (fibrin-polyethylene glycol carriers) that releases SDF-1 for 28 days (mouse model of myocardial infarction) [34]. A recently investigated method of increasing the concentration of SDF-1 is the administration of genetically modified mesenchymal stem cells (MSC) from the bone marrow overexpressing SDF-1 [35]. However, this kind of modifications may pose a safety risk as it may result in a sustained increase in SDF-1α expression with unknown consequences. 

The key issue is to develop methods for the safe and efficient transfer of SDF-1 that will ensure the sustained release of this factor into the heart muscle at concentrations necessary to maintain a chemotactic gradient sufficient to recruit circulating stem cells. Previous studies suggest that it is likely to be achieved by a controlled release system from microspheres after intra-pericardial administration [36]. 

In our latest research, we developed biodegradable microspheres loaded with SDF-1α, which was a novel approach. The microspheres obtained from poly(L-lactide/glycolide/trimethylene carbonate) were characterized by a regular spherical shape and smooth surface and provided prolonged release of SDF-1α (only 40% of SDF-1α was released within 21 days) [18]. This trait allows them to attract cells, which is also valuable for protein functionality on further steps. This is important because the results of the release of SDF-1α in vitro, published so far, mostly indicated a much faster elution from the delivery systems [37,38,39,40]. Another advantageous feature of the microspheres presented in this study is the high encapsulation efficiency of the SDF-1α (67%), which resulted in a much higher concentration of the chemokine in the microspheres (1.4% *w*/*w*) [18], compared to the microparticles reported thus far [41]. 

Moreover, we were the first to be able to significantly extend the presence of SDF-1α in infarcted tissue. To demonstrate this, we injected intrapericardial microspheres 14 days after MI, where levels of endogenous SDF-1α are known to be already low. Our results showed high levels of SDF-1α up to 15 weeks after the myocardial infarction (Figure 3B).

Further, our recent in vitro studies [18] as well as in vivo mouse model studies [42], confirmed that SDF-1α is required for stem cell recruitment to the heart after MI, and that forced overexpression of SDF-1α can enhance stem cell migration and recovery after infarction.

The expression of VCAM and ICAM genes, which are important in the recruitment of stem cells, is known to be increased after myocardial infarction [42,43,44,45]. This is also confirmed by our results, where we observed an increase in the VCAM and ICAM expression on the 3rd day after MI (Figure 4). We have also proven that by releasing SDF-1α from microspheres, this effect can be significantly prolonged, and the level of expression of the genes responsible for the activation of stem cells can be increased (Figure 4).

Overall, our study reveals that adequate delivery of SDF-1α is sufficient to induce the homing of endogenous stem cells to the damaged heart and promote heart repair. Our experiment showed that cytokines released in the pericardium penetrate the heart muscle and the infarct zone, stimulating genes there that can accelerate the regeneration of the heart muscle damaged by acute ischemia. Due to microspheres having such biodegradability, biocompatibility, and lack of toxicity for treated cells, they are desired candidates as delivery tools. Our designed microspheres did not reveal immunogenicity, which is important in term of their further use in clinical trials. Moreover, to the best of our knowledge, this is the first microparticulate delivery system of SDF-1α analyzed in vivo on a large animal model (pigs). Most of the studies conducted so far on polymeric carriers of SDF-1α have been conducted on mice [46,47] or rats [48,49]. It is particularly important that our work on the SDF-1α cytokine and its results can serve as a model for analogous testing of other cytokines and stem cells triggered by these cytokines. This therapy presents the opportunity to overcome limitations such as the low efficacy of nesting of the stem cells in the myocardium.

## 5. Conclusions

The main limitations of stem cell therapy in order to impact heart regeneration are short-term and ineffective stem cell implantation. We believe that the use of a novel treatment methods based on the additional intrapericardial release of a chemotactic factor essential for heart regeneration allows one to overcome this obstacle. Validation of this method in a large animal model will allow us not only to implement endocardial therapy in ischemic cardiomyopathy in the future, but will also provide a platform for the delivery of drugs and biological substances to the pericardium for the treatment of other diseases, namely bacterial and autoimmune pericarditis or neoplastic diseases, to an extent.

Further research is needed that includes the identification of factors that affect the efficiency of SDF-1α-released microspheres and their delivery to the pericardial sac, and their influence on the nesting of stem cells at the target destination.

## Figures and Tables

**Figure 1 biomedicines-11-00343-f001:**
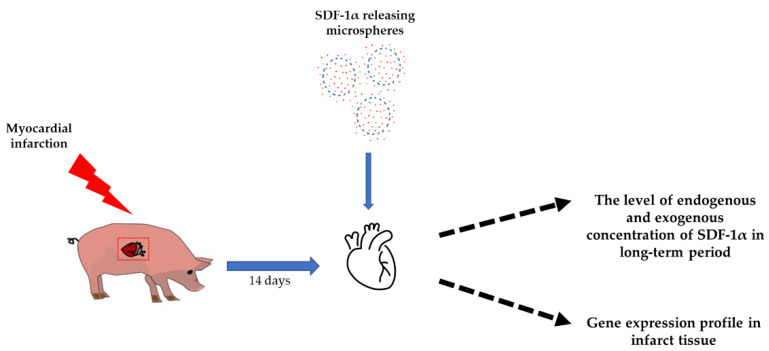
Study design. Fourteen days after myocardial infarction, SDF-1α released from microspheres were transferred to the pericardium sac. The effect of SDF-1α was investigated using ELISA test, measuring the level of endogenous and exogenous concentrations of SDF-1α over a long-term period, and RT-qPCR, examining the gene expression profile in the infarct tissue.

**Figure 2 biomedicines-11-00343-f002:**
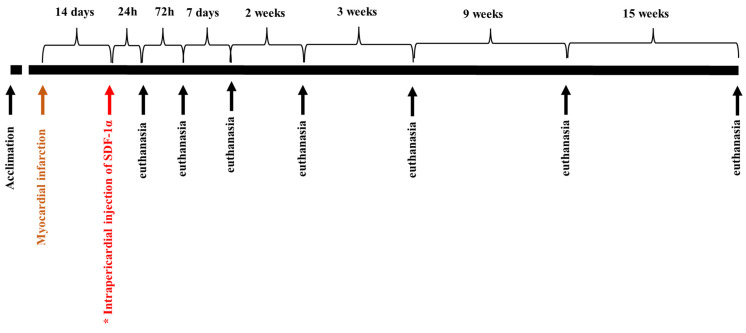
The scheme of the experiment. In the study group (*), SDF-1α-released microspheres were delivered to the pericardial sac 14 days after the myocardial infarction. Control animals did not receive microspheres after MI. The animals were euthanized 24 h, 72 h, 7 days, 14 days, 3 weeks, 9 weeks, and 15 weeks after microspheres delivery.

**Figure 3 biomedicines-11-00343-f003:**
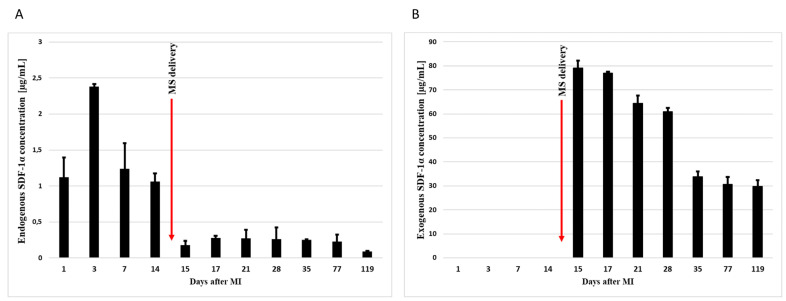
Concentration of endogenous (**A**) and exogenous (**B**) SDF-1α in infarct tissue. Concentrations were measured by ELISA 1, 3, 7, 14, 15, 17, 21, 28, 35, 77, and 119 days after the myocardial infarction. In addition, 14 days after MI, SDF-1α-released microspheres were transferred to the pericardium sac. The bars represent the means ± SD (*n* = 3).

**Figure 4 biomedicines-11-00343-f004:**
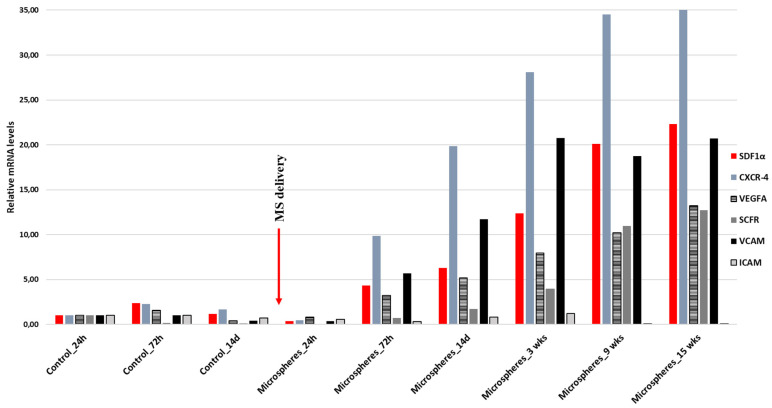
RT-qPCR expression analysis of SDF-1α (stromal cell-derived factor 1), CXCR-4 (C-X-C chemokine receptor type 4), VEGFA (vascular endothelial growth factor A), SCF (stem cell factor), and VCAM (vascular cell adhesion molecule) after myocardial infarction (MI). The study group represents animals that, two weeks after the myocardial infarction, received into the pericardial sac, microspheres releasing SDF-1α. The control group represents animals that did not receive microspheres. Gene expression was evaluated at different time points: for the control group, 24 h, 72 h, and 14 days after MI, and for the study group, 24 h, 72 h, 14 days, 3 weeks, 9 weeks, and 15 weeks after delivery of SDF-1α-releasing microspheres. Values are mean ± SD (*n* = 3).

**Table 1 biomedicines-11-00343-t001:** Study procedures’ flowchart.

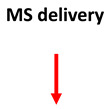											
	Control Group			Study Group							
Animals number	*n* = 3	*n* = 3	*n* = 3	*n* = 3	*n* = 3	*n* = 3	*n* = 3	*n* = 3	*n* = 3	*n* = 3	*n* = 3
Days after MI	24 h	72 h	7 d	14 d	15 d	17 d	21 d	28 d	35 d	77 d	119 d
Days after MS delivery					24 h	72 h	7 d	14 d	3 wks	9 wks	15 wks
ELISA test	x	x	x	x	x	x	x	x	x	x	x
RT-PCR	x	x		x	x	x		x	x	x	x

## Data Availability

Not applicable.

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
