# Peer review of "SDF-1α-Releasing Microspheres Effectively Extend Stem Cell Homing after Myocardial Infarction"

_biomedicines, 2023, doi:10.3390/biomedicines11020343_

Round 1
Reviewer 1 Report
- the English of the paper should be revised by a native English speaker in order to improve the readability of the text. It is really difficult to follow the text
- Results section of the abstract should include more numerical data in order to make it outstanding
Author Response
Reviewer 1
Comment 1: The English of the paper should be revised by a native English speaker in order to improve the readability of the text. It is really difficult to follow the text
Response to Comment 1: We have improved the English throughout the manuscript.
Comment 2: Results section of the abstract should include more numerical data in order to make it outstanding
Response to Comment 2: The abstract section has been supplemented with detailed results. Please, see enriched section “Abstract” in the revised manuscript.
Reviewer 2 Report
In this paper, Bajdak-Rusinek K. et al analyzed in vivo on a large animal model (pigs) a new method to eliminate the basic limitations of cell therapy, consisting of a delivery system of SDF-1α.
There are some major and minor comments to be addressed.
Major comments
1) Extensive editing of the English language and style is required from the abstract to the conclusions
2) On page 2, lines 53-54, you should clarify better this sentence “Therefore, one of the most promising strategies for expanding therapeutic panel in the treatment of cardiovascular diseases is stem cell therapy.”, in respect to the previous strategy.
3) In the methods section you should clarify the time period of your study, the type of pigs you studied and how did you calculate the number of them.
4) On page 3, line 92, for the sentence “a recognized model of myocardial infarction” you should put a reference.
5) You should clarify why you delivered the microsphere 14 days after the AMI.
6) The results section should be rewritten. In this section, you are still talking about methods. You should rewrite this, inserting data (numbers) of your results. Not just referring to the figures. Furthermore, you should insert at least one main table of results!
7) The discussion section should be concise and straight to the point. In the majority of the section, you did a sort of review. Even if there is something you could insert in the introduction (only a few sentences), in the discussion you should insert comments on each important finding, in light of what is already known in the literature and what you will want to do. As you wrote, the reader could lose your study and concentrate on a literature review. Maybe it could be an idea to write in the future, after the publication of your work, a review.
8) A limitation section should be inserted.
Minor comments
1) double-check the spacing between words throughout the text, also in the captions of the figures
2) in the caption of the figures, you should spell the abbreviations of the genes (SDF-1α, CXCR-4, VEGF, SCF, and VCAM)
3) figure 3 should be improved in quality.
Author Response
Reviewer 2
General Comment: In this paper, Bajdak-Rusinek K. et al analyzed in vivo on a large animal model (pigs) a new method to eliminate the basic limitations of cell therapy, consisting of a delivery system of SDF-1α.
There are some major and minor comments to be addressed.
Response to General Comment: We very much appreciate the overall positive evaluation of our review. We are grateful for valuable comments.
MAJOR COMMENTS
Comment 1: Extensive editing of the English language and style is required from the abstract to the conclusions
Response to Comment 1: We have improved the English throughout the manuscript.
Comment 2: On page 2, lines 53-54, you should clarify better this sentence “Therefore, one of the most promising strategies for expanding therapeutic panel in the treatment of cardiovascular diseases is stem cell therapy.”, in respect to the previous strategy.
Response to Comment 2: The sentence has been changed.
Comment 3: In the methods section you should clarify the time period of your study, the type of pigs you studied and how did you calculate the number of them .
Response to Comment 3: We have supplemented method section and added the type of pigs that have been used for experiments. Additionally, we have added study procedures’ flowchart (Table 1). Please see enriched section Methods (subsection: 2.1. Animal Studies).
Table 1. Study procedures’ flowchart.
Comment 4: On page 3, line 92, for the sentence “a recognized model of myocardial infarction” you should put a reference .
Response to Comment 4: We have completed the literature where swine model of myocardial infarction has been used: Konarski Ł, Dębiński M, Kolarczyk-Haczyk A, Jelonek M, Kondys M, Buszman P. Aborted myocardial infarction in patients with ST-segment elevation myocardial infarction treated with mechanical reperfusion. Kardiol Pol. 2021 Jan 25;79(1):39-45. doi: 10.33963/KP.15650. Epub 2020 Oct 16. PMID: 33078920.
Comment 5: You should clarify why you delivered the microsphere 14 days after the AMI .
Response to Comment 5: Endogenous expression of SDF-1 after myocardial infarction decreases after about 4-7 days. To demonstrate the effect of exogenous SDF-1 released from the microspheres, we had to maintain a time interval after MI where expression of endogenous SDF-1 is no longer present. Hence the administration of microspheres 14 days after MI.
This issue has been clarified in the discussion.
Comment 6: The results section should be rewritten. In this section, you are still talking about methods. You should rewrite this, inserting data (numbers) of your results. Not just referring to the figures. Furthermore, you should insert at least one main table of results!
Response to Comment 6: The results section has been rewritten as suggested.
Comment 7: The discussion section should be concise and straight to the point. In the majority of the section, you did a sort of review. Even if there is something you could insert in the introduction (only a few sentences), in the discussion you should insert comments on each important finding, in light of what is already known in the literature and what you will want to do. As you wrote, the reader could lose your study and concentrate on a literature review. Maybe it could be an idea to write in the future, after the publication of your work, a review .
Response to Comment 7: We are grateful for this valuable comment. The discussion section has been corrected according to the suggestions. The obtained results have been discussed in the context of the available researches. Please see enriched section “Discussion” in the revised manuscript.
Comment 8: A limitation section should be inserted.
Response to Comment 8: The limitation section has been inserted. Please see enriched section “Conclusions” in the revised manuscript.
MINOR COMMENTS
Comment 1: Double-check the spacing between words throughout the text, also in the captions of the figures.
Response to Comment 1: This has been corrected accordingly throughout the revised manuscript.
Comment 2: In the caption of the figures, you should spell the abbreviations of the genes (SDF-1α, CXCR-4, VEGF, SCF, and VCAM).
Response to Comment 2: This has been corrected accordingly.
Comment 3: Figure 3 should be improved in quality.
Response to Comment 3: This has been corrected accordingly.

Reviewer 3 Report
Summary: In the current manuscript, the authors developed SDF-1α releasing biodegradable microspheres with the purpose of increasing the recruitment of stem cells to the ischemic cardiac tissue and the results show that the novel therapy method based on the targeted release of a chemotactic factor essential for cardiac tissue regeneration possess a great clinical potential that can help improving the life of patients who have suffered from a heart attack. However, despite the positive results, some minor points should be addressed ahead of publication.
Therefore, in order to improve their work the authors should consider the following recommendations:
1. The “Abstract” section should be reconsider and rephrased so that the methodology and the obtained results could be better understood by readers.
2. The manuscript contains several phrasing and writing mistakes, therefore a major correction in terms of the written English is recommended. Here are several examples: “although the conventional pharmacological and interventional methods for the treatment of IHD presents with relative success” (lines 18-19); “proven to by the natural process” (line 22); “were administered presented to be a prospective” (line 25); “are at the forefront of cardiovascular diseases and presents as a serious problem” (lines 38-39); “on the stem cells homing the bone marrow and circulating in the blood” (line 212); “investigations undertaken the team” (line 223); “is one of the prognostic factors” (line 241); “the microspheres obtained from poly(L-lactide/glycolide/trimethylene carbonate) characterized regular spherical shape” (lines 266-267); “this trait allows to cell attracting that is also valuable for protein..” (line 269); “so far the experience includes” (line 281); “they are perfect candidates as delivery tool”(lines 301-302); etc.
3. Please expand the abbreviations at their first use within the manuscript, e.g. SDF-1α; PVA; CXCR-4; VEGFA; VCAM; SCF; DPBS.
4. The title of subsection 3.1. “SDF-1α is present in the myocardium following intrapericardial administration of the microspheres” should be reconsidered and changed since this subsection presents the results obtained from the ELISA assessment. Please rephrase the title in such a way that reflects its content.
5. Same concern regarding subsection 3.2. where the title should reflect the results presented there. Please revise..
6. In the “Animal experimental design” subsection the authors stated that the pigs were euthanized after 24h, 72h, 14 days, 3 weeks and 9 weeks but the ELISA assay was performed at 1, 3, 7, 14, 15, 17, 35, 77, 98 and 119 days. How is that possible? Shouldn’t the quantification of SDF-1α take place at the same experimental time points as the euthanasia and RT-PCR?
7. Please further expand the results interpretation obtained after performing the ELISA assay in subsection 3.1.
Author Response
Reviewer 3
General Comment: Summary: In the current manuscript, the authors developed SDF-1α releasing biodegradable microspheres with the purpose of increasing the recruitment of stem cells to the ischemic cardiac tissue and the results show that the novel therapy method based on the targeted release of a chemotactic factor essential for cardiac tissue regeneration possess a great clinical potential that can help improving the life of patients who have suffered from a heart attack. However, despite the positive results, some minor points should be addressed ahead of publication.
Response to General Comment: We very much appreciate the overall positive evaluation of our review. We are grateful for valuable comments.
Comment 1: The “Abstract” section should be reconsider and rephrased so that the methodology and the obtained results could be better understood by readers .
Response to Comment 1: The abstract section has been rephrased and supplemented with detailed results. Please, see enriched section “Abstract” in the revised manuscript.
Comment 2: The manuscript contains several phrasing and writing mistakes, therefore a major correction in terms of the written English is recommended. Here are several examples: “although the conventional pharmacological and interventional methods for the treatment of IHD presents with relative success” (lines 18-19); “proven to by the natural process” (line 22); “were administered presented to be a prospective” (line 25); “are at the forefront of cardiovascular diseases and presents as a serious problem” (lines 38-39); “on the stem cells homing the bone marrow and circulating in the blood” (line 212); “investigations undertaken the team” (line 223); “is one of the prognostic factors” (line 241); “the microspheres obtained from poly(L-lactide/glycolide/trimethylene carbonate) characterized regular spherical shape” (lines 266-267); “this trait allows to cell attracting that is also valuable for protein..” (line 269); “so far the experience includes” (line 281); “they are perfect candidates as delivery tool”(lines 301-302); etc.
Response to Comment 2: We have improved the English throughout the manuscript.
Comment 3: Please expand the abbreviations at their first use within the manuscript, e.g. SDF-1α; PVA; CXCR-4; VEGFA; VCAM; SCF; DPBS .
Response to Comment 3: This has been corrected accordingly.
Comment 4: The title of subsection 3.1. “SDF-1α is present in the myocardium following intrapericardial administration of the microspheres” should be reconsidered and changed since this subsection presents the results obtained from the ELISA assessment. Please rephrase the title in such a way that reflects its content .
Response to Comment 4: Title of subsection 3.1. has been changed.
Comment 5: Same concern regarding subsection 3.2. where the title should reflect the results presented there. Please revise.
Response to Comment 5: Title of subsection 3.2. has been changed.
Comment 6: In the “Animal experimental design” subsection the authors stated that the pigs were euthanized after 24h, 72h, 14 days, 3 weeks and 9 weeks but the ELISA assay was performed at 1, 3, 7, 14, 15, 17, 35, 77, 98 and 119 days. How is that possible? Shouldn’t the quantification of SDF-1α take place at the same experimental time points as the euthanasia and RT-PCR?
Response to Comment 6:
Animal experimental design was rewritten because indeed times could be confusing. In addition, we standardized the times for ELISA and RT-qPCR. Unfortunately we do not have the results of gene expression for 7 days after the heart attack and delivery of the microspheres. We hope that the results will nevertheless be accepted. Below is also a table (Table 1) showing how many animals and for what experiments they were used, along with the days of individual analyses.
Table 1. Study procedures’ flowchart.
Comment 7: Please further expand the results interpretation obtained after performing the ELISA assay in subsection 3.1 .
Response to Comment 7: The interpretation of the results obtained after the ELISA test has been expanded in subsection 3.1.

Round 2
Reviewer 1 Report
authors well addressed my previous comments. The paper improved very much
Reviewer 2 Report
None